# LANGUAGE MODEL PRE-TRAINING WITH LINGUISTICALLY MOTIVATED CURRICULUM LEARNING

## ABSTRACT

Pre-training serves as a foundation of recent NLP models, where language modeling task is performed over large texts. It has been shown that data affects the quality of pre-training, and curriculum has been investigated regarding sequence length. We consider a linguistic perspective in the curriculum, where frequent words are learned first and rare words last. This is achieved by replacing hierarchical phrases that contain infrequent words by their constituent labels. By such syntactic substitutions, a curriculum can be made by gradually introducing words with decreasing frequency levels. Without modifying model architectures or introducing external computational overhead, our data-centric method gives better performances over vanilla BERT on various downstream benchmarks.

## 1 INTRODUCTION

Pre-trained language models (PLM) have gained much attention and achieved strong results in various NLP tasks (Devlin et al., 2019; Radford et al., 2019; Brown et al., 2020). Based on self-supervised learning objectives such as causal language modeling (Peters et al., 2018; Radford et al., 2019), masked language modeling (Devlin et al., 2019), and text-to-text generation (Lewis et al., 2020; Raffel et al., 2020), PLM can learn task-agnostic transferable features from large-scale unlabeled corpora. It has also been shown that PLM can encode syntactic (Hewitt & Manning, 2019; Goldberg, 2019; Wu et al., 2020), semantic (Tenney et al., 2019; Jawahar et al., 2019), and factual (Petroni et al., 2019; Dai et al., 2022) knowledge.

For improving the representation power of PLM, much research has been done on setting different training objectives (Zhang et al., 2019; Yang et al., 2019; Liu et al., 2019b), modifying model architectures (Dong et al., 2019; Clark et al., 2020; He et al., 2021) and scaling up the parameter count (Shoeybi et al., 2019; Rae et al., 2021; Fedus et al., 2022; Chowdhery et al., 2022). However, relatively less work considers on the way of using pre-training corpus, where most of the methods leverage the raw text as a whole (from millions to billions of tokens) and train for multiple epochs, given sufficient data, the training strategy may have reduced effect.

Recent work has shown the influence of a curriculum for pre-training. Li et al. (2021) propose a sequence length warmup strategy for GPT-2 pre-training, which can improve training stability and efficiency. Similarly, Nagatsuka et al. (2021) split corpus into blocks with specified sizes for BERT pre-training. These methods focus on changing the sequence length instead of the content and emphasize the convergence speed. Beyond text length, there is a more salient discrepancy between the current PLM training and the language learning process of humans. In particular, we only learn limited but the most common and useful words at the beginning, then we grasp some basic syntactic concepts such as part-of-speech, set phrase, and clause, before recognizing a large number of uncommon words via generalization or their specific usages.

Inspired by psycho-linguistic curriculum learning (Elman, 1990; 1993; Bengio et al., 2009), we propose a data-centric approach that progressively pre-trains a language model using a curriculum that involves reconstructed data. An example contrast of the masked language model pre-training and our multi-stage curriculum training is shown in Figure 1. Our curriculum consists of $m$ stages ($m = 2$ in Figure 1(b)), with each having a incrementally larger vocabulary. Specifically, we first use constituent (and part-of-speech) labels from Penn Treebank (Marcus et al., 1993) to replace the lower frequency words, and the model updates using the text composed of the most frequent words and the constituent labels. In this stage, all words are at the same frequency level, and thus

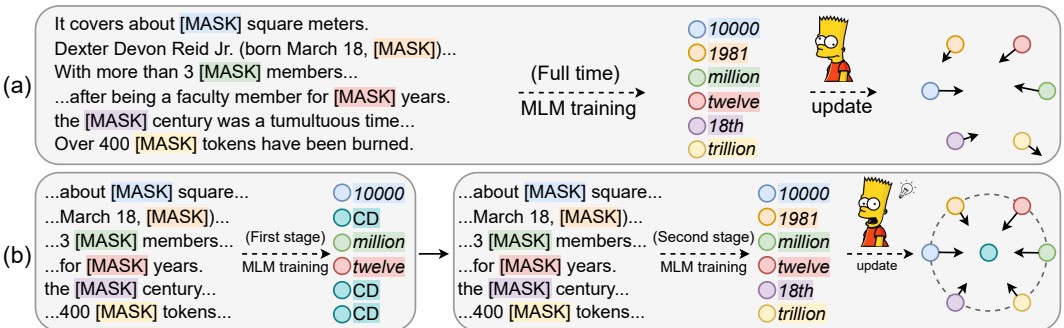

Figure 1: An example of (a) vanilla masked language modeling, and (b) our method using two-stage curriculum training. In the first stage, we replace the original lower frequency target word such as "*trillion*" by a constituent label CD, which stands for the cardinal number. The representations can be better updated in the latter training stage after acquiring the "concept" of CD.

trained equally thoroughly. Then we gradually introduce less frequent words, letting the model further improve based on previously acquired knowledge. During this stage, the previously learned constituent labels can serve as categorical knowledge to guide the learning of infrequent words.

Experimental results using BERT show that our method can improve pre-training, showing better performance across tasks including general language understanding, named entity recognition, question answering, part-of-speech tagging, and parsing. Through empirical analysis, we find that our curriculum training can mitigate the representation degeneration problem (Gao et al., 2019) in PLM, and the injected constituent labels can encode meaningful linguistic features that bridge the word representations across different frequencies. Code and model will be released for further research.

## 2 METHOD

We take BERT (Devlin et al., 2019) as our baseline, which is trained using masked language modeling, one of the most successful self-supervised learning objectives for pre-training (§2.1). Our method leverages linguistically motivated curriculum learning based on vanilla masked language modeling (§2.2), with a dedicated data-centric method for stage-wise corpus reconstruction (§2.3).

### 2.1 MASKED LANGUAGE MODELING

Masked language modeling (MLM) aims to predict the original target word $w_i$ through modeling the contextualized representation of a randomly masked word $\widetilde{w}_i$ in its context:

$$\mathcal{L}_{\text{MLM}} = -\sum_i \log P_\theta(w_i|\widetilde{w}_i) = -\sum_i \log \frac{\exp(E(w_i)^\top \widetilde{h}_i)}{\sum_{j=1}^{|V|} \exp(E(w_j)^\top \widetilde{h}_i)}, \quad (1)$$

where $\widetilde{w}_i$ is the masked symbol [MASK] in a context, $\widetilde{h}_i$ is the corresponding contextualized output.

### 2.2 MOTIVATION FOR DATA-CENTRIC CURRICULUM TRAINING

During vanilla MLM pre-training shown in Figure 1(a), the model needs to predict the corresponding word surface independently. Although the more common words such as "*10000*", "*million*", and "*twelve*" could be updated frequently, lower frequency words such as "*1981*", "*18th*", and "*trillion*" may receive far less training signal. The discrepancy in word frequency may do harm to model training in tasks such as text classification and machine translation (Gong et al., 2018). For language modeling, previous studies have shown that lower frequency words are learned poorly (Schick & Schütze, 2020), which can also degenerate the training process for all other words (Yu et al., 2022).

We consider a psycho-linguistically motivated curriculum learning method by proposing two main rules to address the issues: 1) common words first, rare words next, and 2) models are learned with structural constraints, or syntax. To build a curriculum schedule that satisfies the above rules, we inject constituent labels (Marcus et al., 1993) into raw text, replacing different words in different training stages by their corresponding constituent structures.

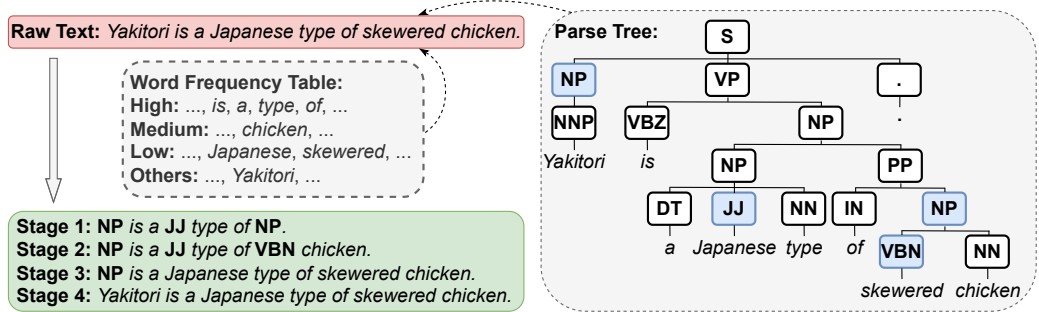

Figure 2: Illustration for reconstructing the sentence "*Yakitori is a Japanese type of skewered chicken.*" in our four stages of curriculum training. At each stage, we use the corresponding constituent labels (colored in blue) to replace the original words according to their overall frequency.

In Figure 1(b), in contrast to the baseline method, we mitigate the influence of infrequent words in the initial stages by predicting the `[MASK]` symbol as a constituent label "CD" (cardinal number) instead of the original word, based on the fact that these words share a unified constituent structure across texts. Then we train our model using the original target after it acquires some "concept" of what CD is. Since the contextualized representation is built for predicting the unified virtual target CD in the first stage, the update of lower frequency word representations could be better guided through such previously learned category knowledge.

## 2.3 RECONSTRUCTING DATA WITH A CONSTITUENCY PARSER

Based on the above motivation, we use a mixup strategy that injects multiple constituent labels into the raw text and progressively incorporates more infrequent words. Our final curriculum includes four stages, where a different number of words in the word frequency table are used in each stage, according to Algorithm 1.

Figure 2 shows an example of injecting constituent labels into a sentence. In the first stage, we only keep the most frequent words ("*is*", "*a*", "*type*", "*of*") while replacing the others by their corresponding constituent labels, these labels are directly used as normal words in the corpus, which can also be randomly masked and predicted. In the second stage, we allow medium frequency words ("*chicken*") to appear together with the most frequent words and remaining labels. In the third stage, we add the low-frequency words ("*Japanese*", "*skewered*") and the data is close to the original format, except for some labels that indicate rare words ("*Yakitori*"). In the last stage, we use the original corpus for training. In

**Algorithm 1** Injecting constituent labels for language model pre-training.

**Input:** Raw text $s_t$, a constituency Parser, a word frequency table TopList
**Output:** Reconstructed text $s_t^\dagger$
1: $s_t^\dagger = [\,]$, Tree = Parser.parse($s_t$)
2: **function** PROCESS(Tree)
3:     **if** Tree has no SubTree **then**
4:         **if** Tree.word in TopList **then**
5:             $s_t^\dagger \leftarrow$ Tree.word
6:         **else**
7:             $s_t^\dagger \leftarrow$ Tree.tag
8:     **else**
9:         **if** all Tree.leaves.word not in TopList **then**
10:             $s_t^\dagger \leftarrow$ Tree.tag
11:         **else**
12:             **for** each SubTree in Tree **do**
13:                 PROCESS(SubTree)

practice, we set the word frequency ranking intervals of ∼0.5K, 0.5K∼3K, 3K∼18K, and 18K∼ as the high, medium, low frequency, and other rare words, respectively. The number of training stages and the frequency intervals are set roughly according to the word distribution and vocabulary size of the embedding table, we leave the optimization of these settings to future work.

To simplify the implementation of our curriculum, we directly use the wordfreq library from (Speer et al., 2018) and ignore the statistics of sub-words (Sennrich et al., 2016; Wu et al., 2016) after tokenization. Thus we can directly process the corpus without considering the distinction of word distribution for different domains, and avoid selecting among tokenizers.

## 3 EXPERIMENTS

### 3.1 PRE-TRAINING

We follow the setup of BERT-base-cased architecture from Devlin et al. (2019). The model is a 12 layers Transformer encoder, with a 768 hidden size and 12 attention heads. English WIKIPEDIA

and the BOOKCORPUS (Zhu et al., 2015) are used as the pre-training data. We train our model with AdamW (Loshchilov & Hutter, 2019) optimizer for 1M steps with a learning rate 1e-4, batch size 256, warmup ratio 0.01, and with mixed precision using 8×32GB V100 GPUs. Following the recipe from Liu et al. (2019c) and Izsak et al. (2021), we do not use the next sentence prediction objective.

For offline data reconstruction, we use the Benepar (Kitaev & Klein, 2018) for parsing. In each of the first three stages, we train our model using the reconstructed corpus for 200K steps (*i.e.*, a total of 600K steps). Then we use the raw corpus for the 600K∼1M training steps. Since we add some constituent labels such as NP, VP, and JJ (see the full list in Appendix A) in the text, we enlarge the embedding table by treating them as normal tokens, thus making our vocabulary size slightly larger (from 28,996 to 29,051)[1]. The 55 externally added embeddings can be discarded after pre-training.

## 3.2 DOWNSTREAM TASKS AND DATASET

We evaluate on general tasks including natural language understanding, named entity recognition, and question answering. Since our method uses text mixed with syntax-related labels during pre-training, we also evaluate on syntax-related tasks such as part-of-speech tagging and parsing. Statistics of the datasets are shown in Appendix B.

**GLUE**. The GLUE benchmark (Wang et al., 2019a) is used for evaluating general language understanding, we compare on sub-tasks including MNLI (Williams et al., 2018), QQP (Chen et al., 2018), QNLI (Rajpurkar et al., 2016), SST-2 (Socher et al., 2013), CoLA (Warstadt et al., 2019), STS-B (Cer et al., 2017), MRPC (Dolan & Brockett, 2005), and RTE (Bentivogli et al., 2009).

**Named Entity Recognition**. We use the CoNLL2003 datasets (Tjong Kim Sang & De Meulder, 2003) for named entity recognition, the entity labels include PER, LOC, ORG, and MISC.

**Question Answering**. Two versions of the Stanford Question Answering Dataset (SQuAD) (Rajpurkar et al., 2016; 2018) are used. SQuAD 1.1 aims to predict the text span in the passage. SQuAD 2.0 allows the possibility that no answer exists in the paragraph.

**Part-of-Speech Tagging**. The Wall Street Journal (WSJ) portion of the Penn Treebank (Marcus et al., 1993) is used for POS tagging. We follow Manning (2011) by selecting sections 0-18 as the training set, 19-21 as the development set, and 22-24 as the test set.

**Constituency Parsing**. WSJ is also used for constituency parsing, where we use the standard splits with sections 02-21 as the training set, 22 as the development set, and 23 as the test set.

## 3.3 FINE-TUNING

For constituency parsing, we use the self-attentive encoder (Kitaev & Klein, 2018) and initialize it with different pre-trained models. For sentence-level classification (GLUE), token-level labeling (NER and POS tagging), and span-based question answering (SQuAD), we follow BERT for fine-tuning. Following Liu et al. (2019c) and Lan et al. (2020), we fine-tune STS-B, MRPC, and RTE by starting from a trained MNLI checkpoint. For other tasks, we train separately using single model and single task without data augmentation. Hyperparameter settings are shown in Appendix C.

We compare fine-tuned results using different pre-trained models: 1) The BERT-base-cased checkpoint released by Google, denoted as **BERT**; 2) Our model trained from scratch, denoted as **BERT-reimp**. The main difference between BERT-reimp and BERT is that we do not use the next sentence prediction objective; 3) Our model trained from scratch with curriculum learning, we denote it as **BERT-CL**. The only difference between BERT-CL and BERT-reimp is the training corpus. For our results, we report by averaging five runs with different seeds.

## 3.4 RESULTS

Table 1 shows the results for the GLUE benchmark. We find that BERT-CL consistently outperforms BERT-reimp across all tasks, showing that our curriculum pre-training is useful. Among all models, BERT-CL gives the best averaged results for both development and test set. We find that

---

[1]There is another option to avoid increasing vocabulary size by using the [unused1] to [unused99] tokens for replacing the added constituent labels.

| Model | MNLI (m/mm) Acc. | QQP F1 | QNLI Acc. | SST-2 Acc. | CoLA Mcc. | STS-B Spear. | MRPC F1 | RTE Acc. | *Avg.* |
|---|---|---|---|---|---|---|---|---|---|
| | *(Development Set)* | | | | | | | | |
| BERT | 84.09/83.82 | **87.53** | 90.84 | 92.31 | 57.27 | 88.24 | 89.41 | **65.69** | 82.13 |
| BERT-reimp | 83.58/83.62 | 87.12 | 90.06 | 92.31 | 57.43 | 88.22 | 89.55 | 63.28 | 81.68 |
| BERT-CL | **84.95/84.77** | 87.15 | **91.06** | **93.00** | **62.06** | **88.41** | **91.14** | 65.61 | **83.12** |
| | *(Test Set via Leaderboard)* | | | | | | | | |
| BERT | 84.5/83.6 | 71.1 | 90.1 | 93.6 | 53.3 | **84.9** | 88.3 | **68.4** | 79.7 |
| BERT-reimp | 83.9/82.6 | 71.1 | 90.2 | 93.7 | 51.9 | 82.4 | 88.7 | 62.0 | 78.5 |
| BERT-CL | **85.4/84.4** | **71.2** | **90.6** | **93.9** | **59.8** | 83.9 | **89.5** | 66.5 | **80.5** |

Table 1: Results on GLUE benchmark dev set and test set. The best results are in bold.

| Model | P | R | F1 |
|---|---|---|---|
| BERT | 90.91 | 92.24 | 91.57 |
| BERT-reimp | 90.94 | 91.93 | 91.43 |
| BERT-CL | **91.63**[†] | **92.31**[†] | **91.97**[†] |

Table 2: Results on CoNLL2003 test set.
†: Statistically significant compared BERT-reimp with $p < 0.01$ by t-test.

| Model | SQuAD 1.1 F1/EM | SQuAD 2.0 F1/EM |
|---|---|---|
| BERT | 88.96/81.61 | 75.69/72.51 |
| BERT-reimp | 89.38/82.71 | 78.19/75.20 |
| BERT-CL | **89.87/83.07** | **80.02/77.06** |

Table 3: Results on SQuAD dev set.

the CoLA task shows the largest improvement (+12.2% compared with BERT), which aims to judge the linguistic acceptability of sentences. This task benefits from our curriculum since we offer some syntactic labels during pre-training, nevertheless, our method can also improve the capability for general language understanding tasks such as natural language inference and sentiment analysis.

Table 2 shows the results for NER. BERT-CL gives a 91.97 F1 score, better than both BERT (+0.40) and BERT-reimp (+0.54). Note that the reported result on the CoNLL2003 test set from Devlin et al. (2019) is 92.4 F1 score, however, we did not achieve it with the current library, as discussed in Stanislawek et al. (2019) and Gui et al. (2020). The improvement over BERT and BERT-reimp may come from the fact that the named entity usually forms a `NN` and `NP` structure in the constituency tree and such knowledge can be better acquired in our preliminary curriculum training stages.

Table 3 shows the results for question answering. Our model gives the best results on both datasets (+0.49/+0.36 and +1.83/+1.86 over BERT-reimp). Compared with the sentence-level classification and token-level labeling tasks, question answering is more challenging since it requires understanding both query and passage with long-term dependency. We hypothesize that the improvement is due to that span-based answers usually form common constituent structures such as `NP` and `CD`, where these features are quite useful for answering the majority of questions like "*what...*", "*where...*", and "*how many...*", "*when...*".

Table 4 shows the results for POS tagging. By using the full training set, our model gives better results than BERT and BERT-reimp. Since WSJ POS tagging is a less complicated task with rich training resources, we also 1) use fewer training data with 2% to 75% samples, and 2) fix the model parameters while only training a linear classifier upon the contextualized output, which is also called probing (Conneau et al., 2018; Liu et al., 2019a). We find that BERT-CL still consistently gives better results under low-resource and probing settings.

Table 5 shows the results for constituency parsing. Compared with BERT and BERT-reimp, our model gives an absolute improvement with +0.30 and +0.37 F1 scores, respectively. The advantage of using the complete match metric is more significant, where BERT-CL gives +2.47 and +1.30 absolute improvement, respectively. Note that although we leverage a constituency parser for building reconstructed data mixed with constituent labels, the improvement is non-trivial since we use the general purposed MLM training instead of specifically augmenting training data for parsing.

To analyze the influence of corpus, we evaluate on GLUE and CoNLL2003 NER test set by 1) using only the WIKIPEDIA, and 2) adding CC-NEWS (Hamborg et al., 2017) for pre-training. Results are shown in Table 6. We can see that BERT-CL still outperforms BERT-reimp when changing the pre-training corpus, showing that the curriculum is useful when applied to different corpora.

| | Model | Accuracy by using $p\%$ Training Set | | | | | |
| --- | --- | --- | --- | --- | --- | --- | --- |
| | | $p$=2 | $p$=5 | $p$=25 | $p$=50 | $p$=75 | $p$=100 |
| *Fine-tuning* | BERT | 96.92 | 96.93 | 97.55 | 97.59 | 97.66 | 97.70 |
| | BERT-reimp | 96.71 | 96.96 | 97.60 | 97.61 | 97.66 | 97.71 |
| | BERT-CL | **96.95**[†] | **96.97** | **97.62** | **97.63** | **97.70**[‡] | **97.75**[‡] |
| *Probing* | BERT | 94.73 | 95.15 | 95.92 | 96.00 | 96.05 | 96.06 |
| | BERT-reimp | 94.72 | 95.20 | 95.82 | 95.88 | 95.91 | 95.92 |
| | BERT-CL | **94.96**[†] | **95.34**[†] | **96.02**[†] | **96.07**[†] | **96.12**[†] | **96.17**[†] |

Table 4: Results on WSJ POS tagging test set by model fine-tuning and linear probing. †, ‡: Statistically significant compared BERT-reimp with $p < 0.01$ and $p < 0.05$ by t-test, respectively.

| Model | LR | LP | F1 | CM |
| --- | --- | --- | --- | --- |
| BERT | 95.20 | 95.32 | 95.26 | 52.06 |
| BERT-reimp | 94.98 | 95.41 | 95.19 | 53.23 |
| BERT-CL | **95.54** | **95.58** | **95.56** | **54.53** |

Table 5: Results on WSJ parsing test set. CM means complete matching the constituency tree of the whole sentence.

| Model | GLUE | CoNLL2003 |
| --- | --- | --- |
| (*wikipedia only*) | | |
| BERT-reimp | 78.2 | 91.49 |
| BERT-CL | **79.9** | **91.90** |
| (*wikipedia, bookcorpus, cc-news*) | | |
| BERT-reimp | 78.9 | 91.52 |
| BERT-CL | **80.7** | **91.98** |

Table 6: Results on GLUE and CoNLL2003 by using different pre-training corpus.

To compare with existing work, we reimplement the method from Nagatsuka et al. (2021) which use a four-stage curriculum with increasing block-size (64, 128, 256, and 512) for BERT pre-training. We follow their setting with 250K training steps for each stage and compare it with our method

| Model | GLUE | CoNLL2003 |
| --- | --- | --- |
| Nagatsuka et al. (2021) | 78.2 | 90.91 |
| BERT-CL | **80.5** | **91.97** |

Table 7: Comparison with an existing method that leverages curriculum learning for pre-training.

in Table 7. We find that the length-based curriculum does not help much in downstream tasks and our method significantly performs better on both GLUE and CoNLL2003 NER tasks. This shows that our content-based curriculum is not only closer to the process of language learning of humans, but also more helpful for model training.

To further evaluate the generalizability of our method, we also try our method to 1) different model settings including larger model RoBERTa-large, and the generative-style language model GPT-2; 2) different curriculum training schedule. We find that our method can also generalize to larger model or GPT-2-style causal language model training, and the curriculum schedule can also affect the overall performance. Detailed results and discussion can be found in Appendix D.

## 4 ANALYSIS

We analyze the possible reasons behind the performance advantage of BERT-CL, discussing how data-centric curriculum training helps language model pre-training.

### 4.1 REPRESENTATION DEGENERATION OF LANGUAGE MODEL

Existing work shows the representation degeneration problem of language models (Gao et al., 2019; Ethayarajh, 2019; Wang et al., 2019b; Cai et al., 2020; Biś et al., 2021; Yu et al., 2022), where the word embeddings or contextualized output are highly anisotropic and may limit the representation power. This problem also exists when training static word embeddings (Mu et al., 2018). Figure 3 visualizes the evolution of word embeddings over training iterations using PCA. In the top row, we find that the embeddings of BERT degenerate quickly in the early stage, where the overall distribution falls into a relatively narrow angle. The low-frequency words are significantly separated from others and the overall shape does not change much during all training iterations.

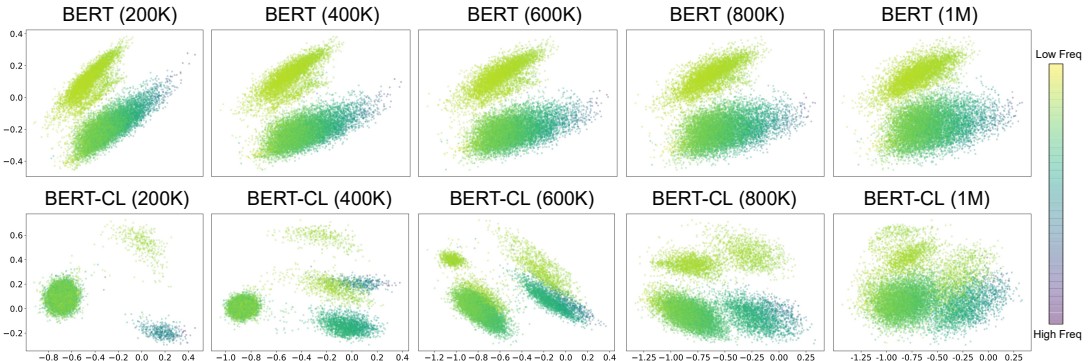

Figure 3: Visualization of word embeddings during pre-training, the numbers in the parentheses denote the training steps. Different colors mean different word frequencies.

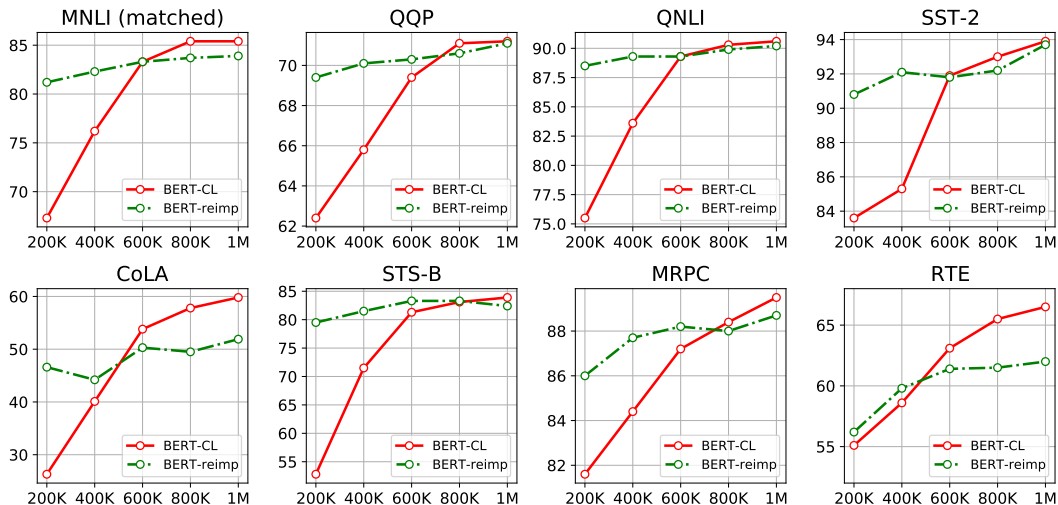

Figure 4: The GLUE benchmark test set results from intermediate checkpoints during pre-training.

The evolution of word distribution in BERT-CL is highly different. Since we incorporate words in different frequency intervals at different stages, they gradually form their clusters stage-by-stage until 600K steps. When using the raw corpus for the last 400K steps of training, we find that the low-frequency words are not represented separately, and different clusters come close to each other. Finally, there is no significant borderline for words with different frequencies, and the overall distribution is also more uniform than BERT. In addition to visualization, we use a measure of isotropy in Mu et al. (2018) and Rajaee & Pilehvar (2021) for evaluation. Details and results are shown in Appendix E, we find that our curriculum training leads to a more isotropic representation space quantitatively.

One of the reasons behind the representation degeneration problem is the frequency discrepancy between words. For example, Gong et al. (2018) find that the word embeddings are heavily biased towards word frequency, proposing an adversarial training method to learn frequency-agnostic representations. Gao et al. (2019) theoretically show that it could be caused by a large number of rarely appeared tokens, and they use a cosine regularization term to enforce normalizing the distribution. Yu et al. (2022) leverage an adaptive gradient gating mechanism for rare tokens training. Although these methods alleviate the degeneration problem and improve performance in pure language modeling or machine translation task, they require additional computational costs and are not used in pre-training. In contrast, we use a data-centric curriculum pre-training approach that introduces the constituent labels and decouples the words with different frequencies by reformulating the corpus.

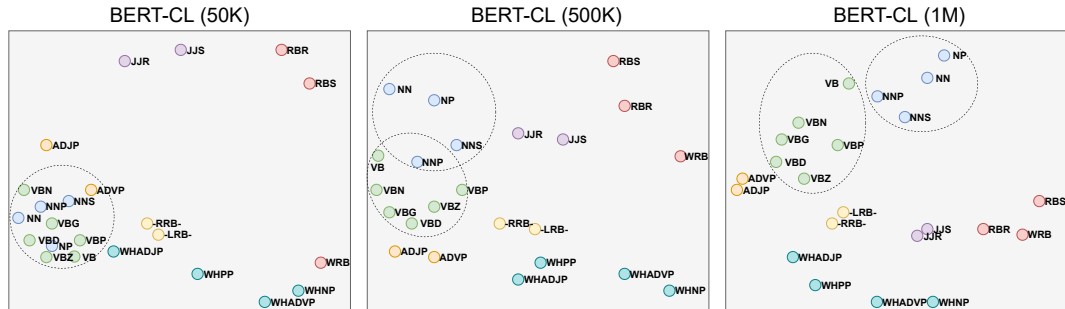

Figure 5: Visualization of the learned constituent label embeddings using t-SNE.

## 4.2 Intermediate Results of Curriculum Training

In Figure 4, we show the GLUE benchmark test set results using different checkpoints during BERT pre-training with and without curriculum training. We find that the BERT model gives a relatively higher performance in the initial stages for almost all tasks. However, the improvement is limited as the training step increases and also inconsistent for tasks such as SST-2, CoLA, STS-B, and MRPC.

For BERT-CL, the performance is lower at the very beginning, but significantly and stably increases then, especially for the first 600K training steps. This shows that curriculum training can boost the capability of the model constantly. We also find that, for MNLI, QNLI, CoLA, and RTE, BERT-CL (600K) already performs on par with or better than BERT (1M). This shows that the reconstructed data can also steer PLM capability for certain tasks with less training cost.

## 4.3 Visualization of Constituent Label Embeddings

The added constituent labels are heavily used during the early pre-training stages for data reconstructing, and their corresponding representations are also updated according to their parameters in the extended embedding table. Although the external added parameters can be discarded during fine-tuning, we are still interested in the role of these constituent labels during curriculum training.

Figure 5 visualizes the 2D distributions from the embeddings of constituent labels. In the very early stages (steps 0∼50K), constituent structures can be learned quickly, where some small meaningful clusters emerge and some relationships are also mined. For example, the groups of nouns and verbs, and the similarity between JJR/JJS and RBR/RBS pairs. The discriminating distribution becomes clearer when the training steps increases, where the nouns and verbs (also the most common constituent structures) are gradually separated from each other. These show that our model could learn some meaningful concepts of these constituent structures during pre-training.

## 4.4 Constituent Labels Serve as Anchors to Bridge Token Learning

In the initial stage, the constituent labels are combined with the most frequent words so that the model can learn some syntax rules, the basic usage of frequent words or terms, and their interactions. Since we only leave the high-frequency words and a bunch of labels, the vocabulary size is much smaller, the training signal received for each token is enriched and also more uniform.

The initial results can offer the fundamental capability for language understanding, we then allow more medium and low frequency words to participate in pre-training, combined with constituent labels and most frequent words that have been well learned. When adding infrequent words to replace the constituent labels, since the words and labels share a similar context, the learned contextualized knowledge from constituent labels can also serve as guidance to the latter learning process of the upcoming words. From this perspective, the curriculum settings of using constituent labels can bridge the gap between words across different frequencies.

Table 8 shows the most similar words to some constituent labels in the trained embedding table. We find that 1) The neighboring words can reflect the fine-grained linguistic characteristics of the constituent labels. For example, the plural number ("*games*", "*states*", "*children*") of NNS, the 3rd person singular present ("*is*", "*does*", "*has*") of VBZ, and the cardinal number ("*five*", "*million*", "*00*") of CD; 2) Different words are captured reasonably according to each constituent label, for

| Lables | High (Top 1~500) | Medium (Top 500~3000) | Low (Top 3000~) |
|---|---|---|---|
| NNS | *games, states, children* | *teeth, victims, units* | *boxers, responses, bands* |
| NP | *him, she, he* | *steps, himself, teeth* | *descent, ghosts, witches* |
| VBZ | *is, does, has* | *becomes, feels, gets* | *receives, saves, serves* |
| WHNP | *who, what, where* | *whom, whatever, whose* | *whoever, wherever, wherein* |
| CD | *five, three, four* | *twenty, ten, hundred* | *fifty, fifteen, twelve* |
|  | *million, two, 00* | *thousand, zero, decade* | *forty, 9th, 8th* |

Table 8: The most similar words to each constituent label according to their dot product.

high, medium, and low frequency intervals. This shows that the constituent label can serve as an anchor or prototype to help model words of different frequency ranges in the curriculum; 3) Phrase-level constituent labels that are usually composed of multiple tokens such as noun phrase of NP and wh-adjective phrase of WHNP are also well encoded with meaningful similar words such as "*him*", "*she*", "*he*", and "*who*", "*what*", "*where*". See Appendix F for more examples.

## 5 RELATED WORK

**Knowledge Enhanced LM**. It has been shown that PLM are capable of encoding syntax and semantic knowledge (Hewitt & Manning, 2019; Tenney et al., 2019; Pérez-Mayos et al., 2021). There are also a line of work explicitly integrating such knowledge to enhance model representation (Lauscher et al., 2020; Sachan et al., 2021; Xu et al., 2021b). In particular, Levine et al. (2020) leverage word sense prediction task into BERT pre-training, Bai et al. (2022) propose hypernym class prediction for causal language modeling. These methods focus on word-level external knowledge stored in WordNet. Instead, we uses the hierarchical syntactic tree to inject word-, phrase- and clause-level knowledge. Moreover, through the underlying syntax structure of texts, our method can tackle more situations during pre-training where words are not in WordNet (*e.g.*, url/email address, new words).

**Curriculum Learning**. Curriculum learning has been extensively studied in a range of tasks (Wang et al., 2022). In natural language processing, Bengio et al. (2009) first show that it can help generalization and speed up the convergence of language modeling. For general-purposed language model pre-training, Campos (2021) defines some sentence difficulty metrics based on sentence length, $n$-gram probability, and part-of-speech diversity for curriculum settings on LSTM-based model. Zhang et al. (2021) group sequences with similar length during pre-training and find that it can help downstream tasks. Nagatsuka et al. (2021) propose progressively increasing the block-size of input text, *i.e.*, using sentences of increasing lengths for pre-training. Li et al. (2021) propose a regularization method for GPT-2 curriculum training, which is also based on sentence length. Unlike these methods, we build a linguistically motivated curriculum based on the learning content.

**Data-centric AI**. Data-centric method become an emerging topic for modern AI systems (Ng, 2021; Hajij et al., 2021; Xu et al., 2021a; Huang et al., 2022; Eyuboglu et al., 2022). The main idea is to use an established model off-the-shelf, but engineer the data for stronger results, including data collection, annotation, augmentation, cleaning, reordering, and deduplicating (Russell et al., 2008; Krishnan et al., 2016; Wei & Zou, 2019; Press et al., 2021; Agrawal et al., 2021; Lee et al., 2022). To better leverage the language model for downstream tasks, there is also a trend to build cloze-style samples for fine-tuning (Schick & Schütze, 2021; Gao et al., 2021b). Reconstructed training data such as prompts or instructions are also being studied for better using large language models in few/zero-shot scenarios (Sanh et al., 2022; Wei et al., 2022; Yuan & Liu, 2022). In this paper, we attempt to reformulate the data using a syntax-guided mixup strategy for language model pre-training.

## 6 CONCLUSION

We investigate curriculum learning for language model pre-training and focus on a purely data-centric method, without setting multiple training tasks, modifying model architecture, or introducing external computational overhead during pre-training. Particularly, we propose a data mixup strategy that injects constituent labels into the text and progressively increases the vocabulary on the corpus from high-frequency to low-frequency words. Experiments on multiple downstream tasks show that our method leads to better performance compared with baselines.

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

# A LIST OF CONSTITUENT LABELS

We injected into the corpus with a total of 55 constituent labels defined in Penn Treebank, including:

Labels = [-LRB-, -RRB-, ADJP, ADVP, CONJP, DT, EX, FRAG, FW, INTJ, JJ, JJR, JJS, LS, LST, NAC, NN, NNP, NNPS, NNS, NP, NX, PDT, POS, PRN, PRP, PRP\$, PRT, QP, RBR, RBS, RP, RRC, SBAR, SBARQ, SINV, SQ, SYM, TOP, UCP, UH, VB, VBD, VBG, VBN, VBP, VBZ, WDT, WHADJP, WHADVP, WHNP, WHPP, WP, WP\$, WRB]

A brief description of these labels can be found in http://surdeanu.cs.arizona.edu/mihai/teaching/ista555-fall13/readings/PennTreebankConstituents.html. After data reconstruction, we treat these labels as normal tokens and enlarge our embedding table, making them involved in the masked language model pre-training:

```
tokenizer_kwarg={"additional_special_tokens":Labels}
tokenizer=AutoTokenizer.from_pretrained(tokenizer_name,**tokenizer_kwarg)
model=AutoModelForMaskedLM.from_pretrained(model_name)
model.resize_token_embeddings(len(tokenizer))
```

# B STATISTICS OF DATASETS

Statistics of the datasets are shown in Table 9.

| Dataset | Task | #Train | #Dev | #Test | #Label |
|---|---|---|---|---|---|
| MNLI | Natural language inference | 393k | 20k | 20k | 3 |
| QQP | Paraphrase | 364k | 40k | 391k | 2 |
| QNLI | Natural language inference | 108k | 5.7k | 5.7k | 2 |
| SST-2 | Sentiment | 67k | 872 | 1.8k | 2 |
| CoLA | Acceptability | 8.5k | 1k | 1k | 2 |
| STS-B | Similarity | 7k | 1.5k | 1.4k | 1 |
| MRPC | Paraphrase | 3.7k | 408 | 1.7k | 2 |
| RTE | Natural language inference | 2.5k | 276 | 3k | 2 |
| CoNLL2003 | Named entity recognition | 14.9k | 3.4k | 3.6k | 8 |
| SQuAD1.1 | Reading comprehension | 87.6k | 10.5k | 9.5k | - |
| SQuAD2.0 | Reading comprehension | 130.3k | 11.9k | 8.9k | - |
| WSJ POS tagging | Part-of-speech tagging | 38.2k | 5.5k | 5.4k | 45 |
| WSJ parsing | Constituency parsing | 39.8k | 1.7k | 2.4k | 52 |

Table 9: Statistics of datasets in our experiments.

# C PARAMETER SETTINGS FOR FINE-TUNING

Fine-tuning parameters for GLUE, NER, SQuAD, POS tagging, and parsing are given in Table 10.

| | GLUE | NER | SQuAD1.1&2.0 |
|---|---|---|---|
| Epochs | {3, 5, 10, 20} | 20 | 2 |
| Learning Rate | {2e-5, 3e-5, 5e-5} | 2e-5 | 3e-5 |
| Batch Size | {16, 32} | 16 | 12 |
| Warmup Ratio | {0.06, 0.1} | - | - |
| | POS Tagging (fine-tuning) | POS Tagging (probing) | Parsing |
| Epochs | 20 | 20 | terminated if no improvement on dev set for 60 epochs |
| Learning Rate | 1e-5 | 2e-3 | 3e-5 |
| Batch Size | 16 | {8, 16} | 32 |
| Warmup Steps | - | - | 160 |

Table 10: Parameter settings for fine-tuning downstream tasks.

# D  MORE EXPERIMENTAL RESULTS

## D.1  DIFFERENT MODEL SETTINGS

For larger model settings, we pre-train RoBERTa-large model (355M parameters) on the same corpus and compared on GLUE tasks. In particular, we use the recipe from Wettig et al. (2022) for efficient pre-training by using 40% masking ratio with 500K training steps.

| Model | MNLI | QQP | QNLI | SST-2 | CoLA | STS-B | MRPC | RTE | *Avg.* |
|---|---|---|---|---|---|---|---|---|---|
| RoBERTa-large | 83.9/84.9 | 87.8 | 91.5 | 93.2 | 55.7 | 87.4 | 75.7 | 64.3 | 80.4 |
| RoBERTa-large-reimp | 83.3/84.1 | 88.1 | 91.8 | 93.6 | 51.3 | **88.0** | 75.5 | 64.9 | 80.0 |
| RoBERTa-large-CL | **85.5/85.5** | **88.5** | **92.4** | **94.0** | **56.8** | 87.4 | **80.0** | **66.1** | **81.8** |

Table 11: Comparison between larger models with RoBERTa-large setting.

Results on GLUE dev sets are shown in Table 11. Overall, compared with RoBERTa-large-reimp, our method lead to large improvement on CoLA and MRPC, and also giving close performance or minor improvement across other tasks.

Beyond autoencoding-style model like BERT, our method can apply to auto-regressive model like GPT2 where the reconstructed data is used for left-to-right language modeling. Specifically, we use GPT2-base as our backbone and pre-train on the same corpus. We evaluate them on LAMBADA (Paperno et al., 2016), WikiText2 (Merity et al., 2016) and SWAG (Zellers et al., 2018) without any fine-tuning (*i.e.*, zero-shot) using the LM evaluation framework from Gao et al. (2021a).

| Model | LAMBADA | | WikiText2 | | | SWAG |
|---|---|---|---|---|---|---|
| | ppl. | Acc. | ppl. | byte ppl. | bpb | Acc. |
| GPT2 | 40.06 | 32.54 | 37.30 | 1.96 | 0.97 | 53.78 |
| GPT2-reimp | 36.95 | 33.46 | 31.61 | 1.91 | 0.93 | 54.06 |
| GPT2-CL | **32.97** | **35.67** | **30.13** | **1.89** | **0.91** | **54.56** |

Table 12: Comparison between auto-regressive language models with GPT2-base setting. ppl: perplexity, bpb: bits per byte.

Results are shown in Table 12. We find that the results of auto-regressive LMs are still in favor of the proposed technique, where better results such as lower perplexity are achieved.

## D.2  DIFFERENT CURRICULUM TRAINING STEPS SETTINGS

We investigate the different training steps setting in our four-stage curriculum training, results are shown in Table 13.

| Model / CL Settings | MNLI | QQP | QNLI | SST-2 | CoLA | STS-B | MRPC | RTE | *Avg.* |
|---|---|---|---|---|---|---|---|---|---|
| RoBERTa-large-reimp | 83.3/84.1 | 88.1 | 91.8 | 93.6 | 51.3 | 88.0 | 75.5 | 64.9 | 80.0 |
| RoBERTa-large-CL | | | | | | | | | |
| 20-20-20-40 (%) | 85.5/85.5 | 88.5 | 92.4 | 94.0 | 56.8 | 87.4 | 80.0 | 66.1 | 81.8 |
| 10-20-30-40 (%) | 85.5/85.8 | 88.6 | 92.1 | 93.6 | 56.2 | 87.6 | 79.5 | 66.8 | 81.7 |
| 25-25-25-25 (%) | 84.4/84.8 | 87.8 | 91.9 | 93.2 | 56.0 | 86.8 | 77.7 | 67.1 | 81.0 |
| 10-10-10-70 (%) | 84.2/84.6 | 87.9 | 91.6 | 93.6 | 55.2 | 87.0 | 77.5 | 66.1 | 80.8 |

Table 13: Comparison on different four-stage curriculum training schedule.

Overall, the proposed curriculum training can improve the performance stably. We find that different schedule settings affect the results, where a moderate amount of training using both mixup and raw training data is necessary. For example, training with only 30% steps in the first three stages with the mixup data is not much sufficient. In future work, techniques such as self-paced learning (Kumar et al., 2010; Jiang et al., 2015) can also be considered for setting a better schedule.

# E  MEASURE THE ISOTROPY OF REPRESENTATION SPACE

We measure the isotropy of representation space using the metric defined by Mu et al. (2018), which is also used for measuring recent language models (Rajaee & Pilehvar, 2021):

$$I(W) = \frac{\min_{u \in U} Z(u)}{\max_{u \in U} Z(u)}, \tag{2}$$

where $W$ is the set of representation vectors, $U$ is the set of eigenvectors of $W^\top W$, $Z(u)$ is a partition function define in Arora et al. (2016):

$$Z(u) = \sum_{w_i \in W} \exp(u^\top w_i). \tag{3}$$

The perfect isotropic space would have $I(W)$ close to 1. We calculated the $I(W)$ scores for BERT, BERT-reimp, and BERT-CL, the results are shown in Table 14.

|        | BERT    | BERT-reimp | BERT-CL   |
|--------|---------|------------|-----------|
| $I(W)$ | 1.05e-5 | 6.15e-7    | **1.15e-4** |

Table 14: Measuring the isotropy of representation space of different models.

We find that the $I(W)$ score of BERT-reimp is lower that that of BERT, the reason can be that the original BERT leverage multi-task training (masked language modeling and next sentence prediction). Compared with these two models, BERT-CL gives a higher $I(W)$ score of 1.15e-4, showing that curriculum training can lead to a more isotropic representation space.

# F  MORE EXAMPLES FOR THE MOST SIMILAR WORDS

More examples for the most similar words to each constituent label are shown in Table 15.

| Lables | High (Top 1~500) | Medium (Top 500~3000) | Low (Top 3000~) |
|--------|------------------|------------------------|-----------------|
| NN     | *light, service, group* | *present, mark, mission* | *seed, concentration, penalty* |
| NNP    | *from, for, the* | *present, steel, opposition* | *clay, audio, miniature* |
| NNPS   | *others, children, team* | *crown, lights, figures* | *minors, blues, blacks* |
| VB     | *keep, tell, let* | *promote, kill, develop* | *convert, recover, minimize* |
| VBP    | *are, were, am* | *re, ve, themselves* | *traded, overlap, dwell* |
| VBN    | *taken, given, done* | *broken, combined, dropped* | *torn, risen, divided* |
| VBD    | *took, had, could* | *spoke, closed, ran* | *tore, rolled, slid* |
| VBG    | *taking, saying, looking* | *passing, putting, turning* | *advancing, returning, protecting* |
| WHADVP | *where, when, why* | *whenever, whom, till* | *wherein, whereby, wherever* |
| WRB    | *how, where, whether* | *whenever, lets, forgot* | *wherever, whereby, wherein* |
| RBS    | *far, ago, least* | *highest, worst, anywhere* | *hardest, fastest, shortest* |
| RBR    | *less, oh, ago* | *wanna, faster, longer* | *sooner, hotter, warmer* |
| JJS    | *best, least, most* | *worst, highest, largest* | *lowest, deepest, smallest* |
| JJR    | *better, more, less* | *greater, stronger, larger* | *warmer, happier, thicker* |

Table 15: More examples for the most similar words to the constituent labels. NN: noun; NNP: proper noun, singular; NNPS: proper noun, plural; VB: verb; VBP: non-3rd person singular present; VBN: past participle; VBD: past tense; VBG: gerund or present participle; WHADVP: wh-adverb phrase; WRB: wh-adverb; RBS: adverb, superlative; RBR: adverb, comparative; JJS: adjective, superlative; JJR: adjective, comparative.

