# OpenReview forum: "Language Model Pre-training with Linguistically Motivated Curriculum Learning"
_ICLR.cc/2023/Conference — Submitted to ICLR 2023_

### Official Review · Reviewer_VPVJ · 2022-10-23

**Confidence:** 3
**Correctness:** 2
**Technical Novelty And Significance:** 2
**Empirical Novelty And Significance:** 2
**Recommendation:** 6

**Clarity, Quality, Novelty And Reproducibility:**

The paper is easy to read but the overall structure is weak because the motivation, and any claims the paper wants to make do not seem highlighted. Especially, the paper can be improved by more convincing arguments why CL is really needed here.

Although using CL for pretraining LMs is not new, the proposed CL is simple yet original.

**Strength And Weaknesses:**

The motivation of the paper is quite unclear. Specifically, it is difficult to see, in the paper, the reason CL was chosen. It is true that "there is a more salient discrepancy between the current PLM training and the language learning process of humans", but why is closing this gap helpful? What're specific aspects that the paper is looking for when employing CL? One thing I can see here in the paper, is to deal with rare words, but then the original BERT training already has a quite effective mechanisms, which is the tokenization (resulting in quite a small size vocab, 29k tokens).

The idea of employing syntactic tags to replace rare words / phrases is nice. But CL is not the only way to do. In fact, we can do a simple augmentation: randomly replace words / phrases with their syntactical tags during training BERT. By doing that way, trained BERT will also be aware of syntax and phrasal structures. Following this point, the fact that the new BERT outperforms BERT trained with traditional method could be due to the fact that the new BERT is more aware of syntax, rather than anything related to word frequency.

The experiments do not reflect fully the intention of the paper, which is targeting LM in general. In fact, the paper demonstrates on only one LM which is BERT-based-cased.

The analysis in section 4.1 is unconvincing. In Fig 3, we can see that for both BERT 1M and BERT-CL 1M, there is a rare-word cluster that is away from the rest, and mixture of rare-word and frequent-word clusters. It is difficult to see why BERT-CL mitigates the problem of representation degeneration. Should it be clear if methods of Gong 2018 or Gao 2019 are used for a more thoughtful analysis?





**Summary Of The Paper:**

The paper proposes a curriculum learning (CL) method to train BERT, which is treated as blackbox (no need to modify the original implementation). The CL method consists of several stages, each of which is to replace rare words / phrases with syntactical tags (e.g. 1000 with CD). By doing that, BERT will only need to learn frequent words first, then rare words later. The paper shows that CL method helps mitigate the representation degeneration problem of BERT (i.e. BERT is biased towards frequent words). Also, it demonstrates that BERT trained with the CL outperforms the traditional training way on several down-streaming tasks.

**Summary Of The Review:**

I like the idea of the paper, but the paper should be improved to meet the conf. standard:
1. the motivation is unclear,
2. the experiment doesn't support the claim (only BERT-base-cased is demonstrated rather than LM in general)
3. the analysis of mitigating the degeneration problem is unconvincing

---

> ### Author Response · Authors · 2022-11-13
> **Responses [4.1]-[4.3] to Reviewer VPVJ**
>
> We thank you for your thoughtful comments. We see that your main concern is the motivation of this work. Below we address your specific points:
>
> >[4.1] The motivation of the paper is quite unclear. Specifically, it is difficult to see, in the paper, the reason CL was chosen. It is true that "there is a more salient discrepancy between the current PLM training and the language learning process of humans", but why is closing this gap helpful? What're specific aspects that the paper is looking for when employing CL?
>
> We've tried to clarify the motivation or advantage of the proposed CL settings, especially in section1. p4, section 2.2, and Fig.1. Here we conclude the key points below:
>
> Taking the 2-stage training in Fig. 1 for example: In the first stage, we hope to replace the less frequent words like "*1981*", "*18th*", and "*trillion*" with the common label "CD", the advantages are two-fold: 1) The overall word vocabulary is much smaller and balanced compared with the original long-tailed distribution, 2) Yu et al. (2022) have shown that infrequent words can also degenerate the training process for other words. Thus the model could learn the most fundamental capability for most common words and simple syntax knowledge without being influenced by other noise or rare words (this is also mentioned in section 1. p4 and section 4.4 p1, p2). We hope such curriculum training can lead to a less biased starting point for the later training (where BERT space is strongly degenerated at an early stage, as shown in section 4.1, fig.3), and such training can also get some basic language understanding capability (where curriculum training with less training cost also show strong results in some tasks, as shown in section 4.2, fig.4).
>
> In the second stage, we recover these labels to their original form, i.e., words. First, this stage is necessary, since we aim to derive a general pre-trained language model, less frequency words would not be learned without such "curriculum" of recovery, which will definitely hurt the overall performance (where significant improvement still exists in some tasks, as shown in Fig.4). Second, during the later training stages, the category knowledge that captured by the consistent labels in the previous curriculum provides guidance for the upcoming less frequent words (as discussed in section 1. p4 and section 4.4, p1, p2), and such label knowledge can serve as "anchor" or prototype to words with different frequency (section 4.4, Table 8).
>
> Overall, the CL setting is inspired by the language learning process of humans where simple words and syntax are learned first and rare words next via their specific usages. More importantly, our analysis and results confirm our motivation.
>
> >[4.2] One thing I can see here in the paper, is to deal with rare words, but then the original BERT training already has a quite effective mechanisms, which is the tokenization (resulting in quite a small size vocab, 29k tokens).
>
> First, as we mentioned above, our goal is **NOT** only to "deal with rare words", but also to train a overall better pre-trained LM. Second, a tokenizer such as BPE in current LM offers a suitable vocabulary size for modeling, however, the frequency bias still exists in these long-tailed 29k tokens when using the tokenized text for training (e.g., more high-frequency subwords emerge). This will be more severe in pre-training on large corpus and that's also a reason why the representation degeneration still exists in LM like BERT, as we mentioned in section 4.1, p3.
>
>
> The tokenization mechanism and our method play different roles in pre-training, the overall idea and goals of our work are also different from the subwords algorithm like Sennrich et al. (2016). In particular, the tokenization algorithm offers a fixed-size vocabulary for modeling, while our method increase the vocabulary in a curriculum from frequent-only to more words on the corpus level.
>
> Sennrich et al. Neural Machine Translation of Rare Words with Subword Units. ACL 2016
>
> >[4.3] The idea of employing syntactic tags to replace rare words / phrases is nice. But CL is not the only way to do. In fact, we can do a simple augmentation: randomly replace words / phrases with their syntactical tags during training BERT. By doing that way, trained BERT will also be aware of syntax and phrasal structures.
>
> Encoding more syntax and phrasal structures knowledge is **NOT** the initial goal of this work, though we find that our model gives better results on syntax-related tasks. Simply randomly replacing words with syntactical tags does not alleviate the word-frequency bias issue, which will limit the representation power for LM. as we've mentioned in section 2.2. The motivation and advantages of CL have also been illustrated in the above response to Comments [4.1].

---

> > ### Author Response · Authors · 2022-11-13
> > **Responses [4.4]-[4.5] to Reviewer VPVJ**
> >
> > >[4.4] The experiments do not reflect fully the intention of the paper, which is targeting LM in general. In fact, the paper demonstrates on only one LM which is BERT-based-cased.
> >
> > See also Comment [1.1] by Reviewer ET24 and Comment [3.3] by Reviewer 71SG. We added experiments for a larger model (RoBERTa-large) and auto-regressive LM (GPT2) pre-training. We hope that these results can demonstrate the generalizability of our method.
> >
> > The detailed settings and results can also be founded in Appendix D.1 in this revision.
> >
> >
> > >[4.5] In Fig 3, we can see that for both BERT 1M and BERT-CL 1M, there is a rare-word cluster that is away from the rest, and mixture of rare-word and frequent-word clusters. It is difficult to see why BERT-CL mitigates the problem of representation degeneration. Should it be clear if methods of Gong 2018 or Gao 2019 are used for a more thoughtful analysis?
> >
> > First, in addition to the last figures, the visualization gives the progress of the variation of vector space, where we aim to show that the degeneration exists at the very beginning stages of BERT and affects the later training stages for all time. That also reveals the reason why we use curriculum training during the early stages. Second, we follow your advice by using the measure of the I(W) score used in Mu et al. (2018) (their eq. 1) to evaluate the isotropy, which is also used for evaluating recent language models as in Rajaee et al. (2021) (Gong 2018 and Gao 2019 also do not provide a detailed quantitative evaluation). The results are listed below and they show that BERT-CL can lead to a more isotropic representation space:
> >
> >
> > | | BERT | BERT-reimp | BERT-CL |
> > |:-------------:|:-------------:|:-------------:|:-------------:|
> > | I(W)      | 1.05e-5|6.15e-7|**1.15e-4**|
> >
> >
> > We've put the details in Appendix E in this revision.
> >
> > Mu et al. Simple and Effective Postprocessing for Word Representations. ICLR 2018
> > Rajaee et al. A Cluster-based Approach for Improving Isotropy in Contextual Embedding Space. ACL 2021

---

> > > ### Comment · Reviewer_VPVJ · 2022-11-24
> > > **reply**
> > >
> > > I would like to thank the authors for the response. The authors did address my concerns about the generalisation of the method (via the performance of different LLMs), and other possibilities that don't require curriculum learning. I thus raise my score to 6, with leaning to accept the paper.

---

### Official Review · Reviewer_71SG · 2022-10-24

**Confidence:** 5
**Correctness:** 2
**Technical Novelty And Significance:** 3
**Empirical Novelty And Significance:** 3
**Recommendation:** 5

**Clarity, Quality, Novelty And Reproducibility:**

This idea of two-stage curriculum training and low-frequency replacement has been introduced by Bai et al in ACL 2022. This ACL paper finds that two-stage curriculum training can help language modeling, by replacing the low-frequency words with labels, training LM first with the replaced text, and finally training the LM with the original text. The method is basically the same. The differences are:
- Bai et al use WordNet's hypernym relation to find the replaced label for the low-frequency tokens, while this paper uses a parser to get the labels for the low-frequency tokens.
- Bai et al train an LM and evaluate the perplexity with WikiText103 and Arxiv, while this paper pretrain an LM and evaluates with downstream tasks.

In summary, in this submission, I did not find any discussion of this ACL paper. I think they are quite similar in terms of the method and read like an extension of Bai et al. Furthermore, even though this submission focus on pretraining, the model size is smaller than Bai et al.
Some claims of originality and contributions need to be rephrased and corrected.

[1] Bai, He, et al. "Better Language Model with Hypernym Class Prediction." Proceedings of the 60th Annual Meeting of the Association for Computational Linguistics (Volume 1: Long Papers). 2022.

[2] Press, Ofir, Noah A. Smith, and Mike Lewis. "Shortformer: Better Language Modeling using Shorter Inputs." Proceedings of the 59th Annual Meeting of the Association for Computational Linguistics and the 11th International Joint Conference on Natural Language Processing (Volume 1: Long Papers). 2021.

**Strength And Weaknesses:**

Strength:

- 1. The idea is intuitive and well-described.
- 2. The results show the improvements clearly and support the method.

Weakness:

- 1. See the originality concern in the next section.
- 2. Whether the proposed method is robust to different hyper-parameters of the curriculum training is not answered. For example, the current setup is 200K steps per stage, and 400K for the final stage. I would suggest showing the impact of these hyperparameters.
- 3. Reported finetuning results should be averaged across 3 or 5 times of finetuning-evaluating. In the current submission, I cannot find details of this part and I assume that the result of each task is from a single run.
- 4. Only experiment with the BERT base, which is quite small (110M) in the context of pretraining in 2022/2023. Even for non-pretrained LM papers, 110M is a quite small model. For example, [1] trains 257M models. [2] trains 261M models.

[1] Bai, He, et al. "Better Language Model with Hypernym Class Prediction." Proceedings of the 60th Annual Meeting of the Association for Computational Linguistics (Volume 1: Long Papers). 2022.

[2] Press, Ofir, Noah A. Smith, and Mike Lewis. "Shortformer: Better Language Modeling using Shorter Inputs." Proceedings of the 59th Annual Meeting of the Association for Computational Linguistics and the 11th International Joint Conference on Natural Language Processing (Volume 1: Long Papers). 2021.

**Summary Of The Paper:**

This paper proposes a curriculum learning for language model pretraining: training with high-frequency tokens first, and low-frequency tokens later. They are 4 stages of pretraining:
- 1. First 3 stages: training with the text whose low-frequency tokens are replaced with their syntactic label, for example, NP for noun, CD for number, etc.
By changing the replacement frequency threshold, the pretrain task difficulty can be controlled. This paper introduces 3 thresholds and split the first 600k training steps into 200k respectively. (the word frequency thresholds are <0.5k, 0.5k~3k, 3k~18k)

- 2. Final stage: training with the unplaced original text. This stage is the final 400k steps of the training.

Experiments show the BERT trained with their proposed method can outperform the BERT trained without it.
Evaluation tasks include the GLUE benchmark, CoNLL03, SQuAD 1.1 and 2.0, and WSJ pos-tagging and constituency parsing.

**Summary Of The Review:**

Overall, I like this idea and am happy to see this idea works on BERT pretraining.

However, this submission misses a key related work that first proposes and uses the same "low-frequency word replacement" curriculum method for LM training. This submission needs to rephrase and correct its claim of originality. For example, the originality is how to replace words with labels (with a parser instead of the wordnet).

It would be great to add experiments to show the different word-replacing method is the key to BERT training. (the syntactic parser v.s. WordNet Hypernym relation)

Also, since this paper tries to show the curriculum training works for BERT, an analysis of the curriculum hyperparameters should be added.

---

> ### Author Response · Authors · 2022-11-13
> **Responses to Reviewer 71SG**
>
> Thank you for the constructive comments! We are also grateful that you remind us of the related works.
>
> >[3.1] Whether the proposed method is robust to different hyper-parameters of the curriculum training is not answered. I would suggest showing the impact of these hyperparameters.
>
> See also Comment [1.2] by Reviewer ET24. We've shown more results for different curriculum schedule settings in the response.
>
> We've also provided the results and discussion in Appendix D.2 in this revision.
>
> >[3.2] Reported finetuning results should be averaged across 3 or 5 times of finetuning-evaluating. In the current submission, I cannot find details of this part and I assume that the result of each task is from a single run.
>
> The results are averaged by multiple runs. In the last sentence of section 3.3, we've mentioned that "For our results, we report by averaging five runs with different seeds.".
>
> >[3.3] Only experiment with the BERT base
>
> See also Comment [1.1] by Reviewer ET24 and Comment [4.4] by Reviewer VPVJ. We added experimental results (please refer to the response to Comment [1.1]) for larger model (RoBERTa-large) and auto-regressive LM (GPT2), which shows that our approach can also be applied to more general situations.
>
> We've put the detailed settings and results in Appendix D.1 in this revision.
>
> >[3.4] However, this submission misses a key related work that first proposes and uses the same "low-frequency word replacement" curriculum method for LM training. This submission needs to rephrase and correct its claim of originality. For example, the originality is how to replace words with labels (with a parser instead of the wordnet).
>
> We've mentioned the work of Press et al. (2021) in section 5 (data-centric methods). For the work of Bai et al. ("Better Language Model with Hypernym Class Prediction." ACL 2022), we agree that discussion and comparison are necessary, and we've added them in section 5 in this revision.
>
> To clarify the originality, we conclude some key differences to the work of Bai et al. (2022) and the advantages of our method below:
> * Instead of the "word-based" substitution, we leverage constituent structures which encode richer and hierarchical syntax knowledge for word-, phrase-, and clause-level representation.
> * Bai et al. only consider *low-frequency* *nouns* with *long hypernym paths* (see their section 3.1). In contrast, our method is more general to other words like verbs and medium-frequency words.
> * Hypernym can not be always founded for words that are not in WordNet (e.g., url&email address, or new word like "*doomscrolling*", "*covid-19*"...). These words have to keep their original form during training as in Bai et al., however, the text can always be parsed via the underlying syntax structure so that these words can still be attached to their constituent labels accordingly. Take the sentence [**PRP** *It* **VBZ** *'s* **JJ** *hard* **TO** *to* **VB** *stop* **VBG** *doomscrolling*] as an example, we can replace "*doomscrolling*" with VBG whether or not it is in WordNet or has a suitable hypernym class.
> * Bai et al. focus on getting lower perplexity for LM, however, we aim to get a better general PLM for downstream tasks. They evaluate on 2 LM datasets while we evaluate on 5 different benchmarks and 13 datasets.
>
> >[3.5] It would be great to add experiments to show the different word-replacing method is the key to BERT training. (the syntactic parser v.s. WordNet Hypernym relation)
>
> Thanks for your suggestion. Given the time limit, we focus on the curriculum training in this work, and a thorough comparison between the syntactic parser and WordNet Hypernym relation may not be provided timely. We'll follow your meaningful advice in our future work.

---

> > ### Comment · Reviewer_71SG · 2022-11-30
> > **Reply**
> >
> > Thanks for your response and sorry I forgot to paste my reply in time. Here are my concerns not been solved:
> >
> > 1. The investigation of the CL is not extensive. For example, how is the proposed method robust to the frequency threshold selection? Will the definition of what tokens belong to the low-frequency group, the middle-frequency group, and the high-frequency group, be the key to the success of the CL training? It is hard for others to follow this work if these questions are not answered.
> >
> > 2. No experiments to support the above comparison between this work and Bai et.al, for example: is phrase level replacement better than word level? Is verb and medium-frequency words replacement necessary?

---

> > > ### Author Response · Authors · 2022-12-09
> > > **Additional Responses [3.6]-[3.7] to Reviewer 71SG**
> > >
> > > > [3.6] The investigation of the CL is not extensive. For example, how is the proposed method robust to the frequency threshold selection? Will the definition of what tokens belong to the low-frequency group, the middle-frequency group, and the high-frequency group, be the key to the success of the CL training?
> > >
> > > The overall reason that we group the vocabulary is that we want our model to learn by stages (not directly to learn the content of the whole corpus). Here we do a comparison that we only use the high-frequency group (no mid and low group), and only the high-frequency and middle-frequency group (no low group), the results are shown below:
> > > |Model|MNLI|QQP|QNLI|SST-2|CoLA|STS-B|MRPC|RTE|Avg.|
> > > |------|------|------|------|------|------|------|------|------|------|
> > > |RoBERTa-large-CL (no mid and low group)|84.0/84.7|87.9|91.5|93.0|55.8|86.3|78.7|64.6|80.8|
> > > |RoBERTa-large-CL (no low group)|84.8/84.3|88.4|91.9|93.4|56.4|87.2|78.8|65.0|81.3|
> > > |RoBERTa-large-CL	|85.5/85.5|88.5|92.4|94.0|56.8|87.4|80.0|66.1|81.8|
> > >
> > > Results show that adding some fine-grained groups improve the performance. We agree that the settings can be further investigated, however, it's also not practical to study the influence of every threshold combination, and the vocabulary size, domain and size of corpus will all have an effect on the optimal choice.
> > >
> > > > [3.7] No experiments to support the above comparison between this work and Bai et.al
> > >
> > > As compared to Bai et al., we compare with two other word replacement method, i.e., hypernym and synonym from WordNet, using the NLTK package:
> > >
> > > |Model|MNLI|QQP|QNLI|SST-2|CoLA|STS-B|MRPC|RTE|Avg.|
> > > |------|------|------|------|------|------|------|------|------|------|
> > > |RoBERTa-large-CL (hym)|83.8/84.1|88.0|91.9|93.2|52.9|87.0|76.7|63.1|80.1|
> > > |RoBERTa-large-CL (sym)|83.7/84.6|88.2|91.8|93.4|53.2|87.2|75.6|64.0|80.2|
> > > |RoBERTa-large-CL (syntax)|85.5/85.5|88.5|92.4|94.0|56.8|87.4|80.0|66.1|81.8|
> > >
> > > We find that the syntactic replacement still performs better, here we conclude two reasons: 1) there are still a large number of words do not have their hypernym or synonym in WordNet, however, the underlying syntactic structure usually exist so that syntactic replacement is more general. 2) To our knowledge, the hypernym or synonym in WordNet is word-level and does not consider contextualized information, which could lead to incorrect labels. In our method, the parser will give a higher accuracy for contextualized tags.
> > >
> > > Overall, we think the hypernym replacement in Bai et al. may do good for perplexity reduction, however, the syntactic tags replacement have its own advantages on training a general-purpose pre-trained LM.

---

### Official Review · Reviewer_JTe8 · 2022-10-24

**Confidence:** 4
**Correctness:** 3
**Technical Novelty And Significance:** 3
**Empirical Novelty And Significance:** 3
**Recommendation:** 5

**Clarity, Quality, Novelty And Reproducibility:**


- This is very unclear in the abstract: "This is achieved by substituting syntactic constituents for rare words with their constituent labels." The same it is much clearer a the end of page 1.

- Section 2.1 could be just a citation.

- The results of BERT (Devlin et al. 2019) are not really necessary given that you are reporting results with your BERT reimplementation.





**Strength And Weaknesses:**

Strengths: their approach is novel and shows consistent results over multiple benchmarks,

Weaknesses: results are very close between models in multiple benchmarks, it would be good to show whether they are within the range of noise, especially models trained on the same data.  Table 4 for example is extremely close.

**Summary Of The Paper:**

This paper improves pre-training of MLM models (e.g. BERT) with a curriculum learning strategy, where not frequent words are substituted with their constituent labels. Rarer words are then introduced in pretraining in subsequent pretraining stages, until all words are substituted back to their original form.  Their approach shows minor improvements across multiple tasks including NER, QA, POS tagging and Parsing.

**Summary Of The Review:**

Some additional qs:

- Have you thought about using simpler synonyms for rare words instead of the constituent approach?

- Have you thought about using sentence with simpler syntactic structure in the initial phases of pretraining?

- Would be good to cite relevant work on syntactic information inherently present in BERT like models: https://aclanthology.org/N19-1419/, https://aclanthology.org/2021.emnlp-main.118.pdf , and potentially apply these probes to see whether they actually improve.

---

> ### Author Response · Authors · 2022-11-13
> **Responses to Reviewer JTe8**
>
> Thank you for your valuable feedback.
> >[2.1] results are very close between models in multiple benchmarks, it would be good to show whether they are within the range of noise, especially models trained on the same data. Table 4 for example is extremely close.
>
> POS tagging is a relatively simple task where high accuracy can be easily achieved. We've followed your advice by adding the results of the significance test for NER and POS tagging (where the improvement over BERT-reimp is less than 1.0 points) in our revision.
> We hope these results can convince you that our method offers a better pre-trained model across tasks.
>
> >[2.2] This is very unclear in the abstract: "This is achieved by substituting syntactic constituents for rare words with their constituent labels."
>
> We've revised the wording in this revision.
> >[2.3] Section 2.1 could be just a citation.
>
> We shorten the content of this part in this revision.
> >[2.4] The results of BERT (Devlin et al. 2019) are not really necessary
>
> We've followed your advice in this revision.
> >[2.5] Have you thought about using simpler synonyms for rare words instead of the constituent approach?
>
> The suggested synonym prediction could be useful to some extent. However, our method is more than just changing "word-based" targets because whole phrases can also be replaced with constituent labels, where richer and hierarchical syntax knowledge is also considered for word-, phrase-, and clause-level representation.
>
> In addition, synonyms may not be founded easily and accurately when tackling raw corpus for pre-training, where noisy, informal, and OOV words often exist. Please also refer to the responses to similar Comments [3.4]-[3.5] by Reviewer 71SG for details. We appreciate your suggestion and added discussion and comparison with such related work in section 5 in this revision.
>
>
> >[2.6] Have you thought about using sentence with simpler syntactic structure in the initial phases of pretraining?
>
> For a fair comparison with the original BERT training, we did not order the corpus, e.g., simpler or shorter sentences first (related work such as Press et al. (2021) has been mentioned in section 5). Actually, we've compared with Nagatsuka et al. (2021) in Table 7, where a increasing maximum sequence length is used for pre-training. However, we find such a length-based curriculum is not much effective for downstream tasks.
>
> Press et al. Better language modeling using shorter inputs. ACL 2021
> Nagatsuka et al. Pre-training a BERT with curriculum learning by increasing block-size of input text. RANLP 2021
>
> >[2.7] Would be good to cite relevant work on syntactic information inherently present in BERT like models: https://aclanthology.org/N19-1419/, https://aclanthology.org/2021.emnlp-main.118.pdf , and potentially apply these probes to see whether they actually improve.
>
> We've conducted similar syntax-related probing experiments in Table 4, where results show that our BERT-CL indeed encodes more syntactic information. We've mentioned the work of Hewitt et al. (2019) in section 1, p1.
>
> In the revision, we followed your advice and organize these related works in section 5.

---

### Official Review · Reviewer_ET24 · 2022-10-25

**Confidence:** 4
**Correctness:** 3
**Technical Novelty And Significance:** 3
**Empirical Novelty And Significance:** 3
**Recommendation:** 6

**Clarity, Quality, Novelty And Reproducibility:**

*clarity: the paper's algorithm is clear, except for how the training corpus is actually parsed (which parser is used)
*quality: the result is solid, but it is lacking some ablations studies to really understand what caused the improvement (the curriculum of vocabulary or the parse tree).
*novelty: seems quite novel to me
*reproducibility: I didn't find how the training corpus is actually parsed, other than that it seems reproducible.

**Strength And Weaknesses:**

*strength:
1. A working curriculum algorithm for vanilla masked language modeling.
2. parse-tree based masking seems novel to me (but I'm not familiar with prior works)

*weakness:
1. the experiment is a bit limiting, as only a bert-Base is tested. It's unclear if this method can scale up to larger model and datasets.
2. The scheduling of the curriculum learning seems a bit arbitrary and not systematically investigated.
3. unclear how the method can generalize to languages where a parse-tree is not available.

**Summary Of The Paper:**

A parse-tree based curriculum learning framework to learn a pre-train model.

**Summary Of The Review:**

The work is novel and has good result on a small model and dataset. The main weakness is the lack of ablation study and testing on more models and datasets.

---

> ### Author Response · Authors · 2022-11-13
> **Responses to Reviewer ET24**
>
> Thank you for your constructive comments.
> >[1.1] It's unclear if this method can scale up to larger model and datasets.
>
> For the datasets, we've conducted experiments by using different corpus for pre-training, as mentioned in section 3.4 (Table 6). Experimental results for the larger model (RoBERTa-large) and auto-regressive LM (GPT2) are listed below, which shows that our approach can also be applied to more general situations.
>
> |Model | MNLI | QQP | QNLI | SST-2 | CoLA | STS-B | MRPC | RTE | Avg. |
> |:-------------|:-------------:|:-------------:|:-------------:|:-------------:|:-------------:|:-------------:|:-------------:|:-------------:|:-------------:|
> | RoBERTa-large      | 83.9/84.9|87.8|91.5|93.2|55.7|87.4|75.7|64.3|80.4|
> | RoBERTa-large-reimp| 83.3/84.1|88.1|91.8|93.6|51.3|**88.0**|75.5|64.9|80.0|
> | RoBERTa-large-CL   | **85.5**/**85.5**|**88.5**|**92.4**|**94.0**|**56.8**|87.4|**80.0**|**66.1**|**81.8**|
>
> |Model | LAMBDA (ppl.) | LAMBDA (Acc.) | WikiText2 (ppl.) | WikiText2 (byte ppl.) | WikiText2 (bpb) | SWAG (Acc.) |
> |:-------------|:-------------:|:-------------:|:-------------:|:-------------:|:-------------:|:-------------:|
> |GPT2|40.06|32.54|37.30|1.96|0.97|53.78|
> |GPT2-reimp|36.95|33.46|31.61|1.91|0.93|54.06|
> |GPT2-CL|**32.97**|**35.67**|**30.13**|**1.89**|**0.91**|**54.56**|
>
> The detailed experimental settings and results are provided in Appendix D.1 in this revision.
>
> >[1.2] The scheduling of the curriculum learning seems a bit arbitrary and not systematically investigated.
>
> We added experimental results by using models with different four-stage curriculum training steps, and the results are listed below, where we find that changing the curriculum training schedule can affect overall performance.
>
>
> |RoBERTa-large-CL | MNLI | QQP | QNLI | SST-2 | CoLA | STS-B | MRPC | RTE | Avg. |
> |:-------------|:-------------:|:-------------:|:-------------:|:-------------:|:-------------:|:-------------:|:-------------:|:-------------:|:-------------:|
> |&emsp; 20-20-20-40 (%)|85.5/85.5|88.5|92.4|94.0|56.8|87.4|80.0|66.1|81.8|
> |&emsp; 10-20-30-40 (%)|85.5/85.8|88.6|92.1|93.6|56.2|87.6|79.5|66.8|81.7|
> |&emsp; 25-25-25-25 (%)|84.4/84.8|87.8|91.9|93.2|56.0|86.8|77.7|67.1|81.0|
> |&emsp; 10-10-10-70 (%)|84.2/84.6|87.9|91.6|93.6|55.2|87.0|77.5|66.1|80.8|
>
> More discussions are provided in Appendix D.2 in this revision.
>
> >[1.3] how the training corpus is actually parsed (which parser is used)
>
> The parser we used is the Berkeley Neural Parser (https://parser.kitaev.io/). We've mentioned that "For offline data reconstruction, we use the Benepar (Kitaev & Klein, 2018) for parsing." in section 3.1.
>
> >[1.4] unclear how the method can generalize to languages where a parse-tree is not available.
>
> In fact, the parser we used also supports multilingual parsing (high accuracy for 11+ languages). For languages that are not supported or parse-tree resources are not available, they cannot be directly applied in our method, and such a study for low-resource parsing is out of the scope of this paper.

---

> > ### Comment · Reviewer_ET24 · 2022-11-24
> > **Thanks for adding experiments**
> >
> > I am now more confident that the method is working for more general language model pre-training tasks after the responses. However, I find that the method is not systematically investigated, which greatly limits the impact of this work. For instance, what if you replace the rare word by a simple rare word token for curriculum learning? Does this give any improvement? As VPVJ mentioned, the curriculum and syntactic tags are coupled well, but it is unclear whether you need the syntactic tags to make curriculum learning work. VPVJ's idea of randomly replace words / phrases with their syntactical tags during training can also help identify whether curriculum or syntactic tags are necessary. Also, since 71SG mentioned a similar related work, it is interesting if Bai et al. can be compared to this work in your experimental setting (which would be useful to see if syntactic tags is actually crucial).
> >
> > I feel like the paper could become very strong if many of the questions are actually discussed, as it gives insight to what components are actually helpful in pre-training. Currently, it is unclear which component is more important, and the result is not useful for scenarios where curriculum or syntactic tags are not ideal. I would be happy to raise my score if these experiments/ ablations are completed.

---

> > > ### Author Response · Authors · 2022-12-09
> > > **Additional Responses [1.5]-[1.6] to Reviewer ET24**
> > >
> > > > [1.5] As VPVJ mentioned, the curriculum and syntactic tags are coupled well, but it is unclear whether you need the syntactic tags to make curriculum learning work. VPVJ's idea of randomly replace words / phrases with their syntactical tags during training can also help identify whether curriculum or syntactic tags are necessary.
> > >
> > > Actually, we've shown in Figure 4 that the curriculum of recovering the original word form is necessary, where the improvement exist for most of the tasks. Otherwise such rare words will not be learned during training. We replace the words/phrases with syntactical tags without CL, i.e., using such data for training all the time. The results are shown below:
> > > |Model|MNLI	|QQP|QNLI|SST-2|CoLA|	STS-B|MRPC|	RTE|	Avg.|
> > > |-------|-------|-------|-------|-------|-------|-------|-------|-------|-------|
> > > |RoBERTa-large|83.9/84.9|87.8|91.5|93.2|55.7|87.4|75.7|64.3|80.4|
> > > |RoBERTa-large-no-CL|82.9/83.1|87.0|89.9|91.4|54.0|85.6|74.3|62.1|78.9|
> > > |RoBERTa-large-CL|85.5/85.5|88.5|92.4|94.0|56.8|87.4|80.0|66.1|81.8|
> > >
> > > > [1.6] it is interesting if Bai et al. can be compared to this work in your experimental setting (which would be useful to see if syntactic tags is actually crucial)
> > >
> > > We compare with two other word replacement method, i.e., hypernym and synonym from WordNet, using the NLTK package:
> > > |Model|MNLI	|QQP|QNLI|SST-2|CoLA|	STS-B|MRPC|	RTE|	Avg.|
> > > |-------|-------|-------|-------|-------|-------|-------|-------|-------|-------|
> > > |RoBERTa-large-CL (hym)|83.8/84.1|88.0|91.9|93.2|52.9|87.0|76.7|63.1|80.1|
> > > |RoBERTa-large-CL (sym)|83.7/84.6|88.2|91.8|93.4|53.2|87.2|75.6|64.0|80.2|
> > > |RoBERTa-large-CL (syntax)|85.5/85.5|88.5|92.4|94.0|56.8|87.4|80.0|66.1|81.8|
> > >
> > > We find that the syntactic replacement still performs better, here we conclude two reason: 1) there are still a large number of words do not have their hypernym or synonym in WordNet, however, the underlying syntactic structure usually exist so that syntactic replacement is more general. 2) To our knowledge, the hypernym or synonym in WordNet is word-level and does not consider contextualized information, which could lead to incorrect labels. In our method, the parser will give a higher accuracy for contextualized tags.
> > >
> > > *Overall, both the CL and syntactic tags play important roles in our method and we would like to add these discussion in the final version of this manuscript.*

---

### Author Response · Authors · 2022-11-13
**General Response to All Reviewers**

We would like to thank all reviewers for your time and valuable feedback. We have posted the responses and uploaded the revised paper, where **the revisions are marked in RED**.

Here we outline the major changes:
* **More experimental results for the larger model and generative-style LM (Reviewer ET24, 71SG, VPVJ)**: In Appendix D.1, we show more experimental results for different model settings, including RoBERTa-large and GPT2, which we hope can illustrate the generalizability of the proposed method.
* **Experiments on different curriculum schedules (Reviewer ET24, 71SG)**: In Appendix D.2, we investigate the influence of different curriculum schedules, where we find a moderate amount of training using both mixup and raw training data is necessary.
* **Quantitative results for measuring representation degeneration (Reviewer VPVJ)**: In Appendix E, we provide quantitative results measuring the isotropy of vector space according to Mu et al. (2018) and Rajaee et al. (2021), which offers a complementary result for the visualization.
* **Citation and discussion on related work (Reviewer JTe8, 71SG)**: In section 5, we add a discussion on "Knowledge enhanced LM", which we hope can clarify the originality of our work.
* **Significance testing results (Reviewer JTe8)**: We add the significance testing in Tables 2 and 4, which we hope can strengthen the results.
* **Some issues in the presentation (Reviewer JTe8)**: We rephrased the sentence in the abstract, reduced some content in section 2.1, and removed the results of BERT-report accordingly.

Many thanks and kind regards,
Authors

---

> ### Author Response · Authors · 2022-11-19
> **We look forward to hearing from you.**
>
> Dear all reviewers,
>
> As a reminder, the discussion session will end in a few hours.
>
> Thank you all again for your time and valuable feedback. We have posted the responses and uploaded the revised paper, which we hope can address your concerns. Please let us know if you have further questions.
>
> Sincerely,
> Authors

---

### Decision · Program_Chairs · 2023-01-20

**Decision:**

Reject

**Justification For Why Not Higher Score:**

The basic contribution is good, but there are too many problems with this submission. See above.

**Justification For Why Not Lower Score:**

n/a

**Metareview: Summary, Strengths And Weaknesses:**

CLARIFICATION

This was a borderline paper. We conducted a virtual meeting with the reviewers. Two reviewers did not take part in the virtual meeting.

SUMMARY

The paper addresses language model pretraining. It takes a
curriculum learning approach, focusing on frequent subwords
first, rare subwords later.  In the early stages of
training, rare subwords are replaced by part-of-speech tags.
Extensive experimentation shows that the new method modestly
(but clearly) outperforms prior work as reimplemented /
rerun by the authors. The strongest results (even compared
to state of the art for models of this size) are for
"syntactic" tasks (and, oddly, for squad). This makes sense
as the model is trained on syntactic signals not available
in prior work.


STRENGTHS

The idea of replacing input words by part-of-speech tags and
thereby training a model with better syntactic capabilities
is good. It is surprising that nobody has done this
before. If that's true (i.e., there is no prior work on
this), then this would be the strongest reason for accepting
the paper.

(This:
https://arxiv.org/abs/2211.05344
is somewhat similar, but was published on 2022-11-11, so
not relevant for this decision.)


WEAKNESSES

This paper:

Bai, He, et al. "Better Language Model with Hypernym
Class Prediction." Proceedings of the 60th Annual Meeting of
the Association for Computational Linguistics (Volume 1:
Long Papers). 2022.

is closely related. Instead of initially replacing words
with POS tags, Bai et al initially replace words with
hypernyms. Similar to the submitted paper, there is a lot of
replacement in the beginning of training and no replacement
at the end.

So in a sense the submitted paper does what Bai et al do,
except that the "abstraction" introduced in the beginning of
pretraining is syntactic, not semantic.

This limits the novelty of the submitted paper. But doing
syntactically what Bai et al did semantically still clearly
is a novel contribution.

The fact that the authors did not cite and discuss Bai et al
does not instill trust in the paper. I would be inclined to
reject it for that reason.

Another issue with the paper is that there are other ways of
implementing the idea of "abstraction" in pretraining. As
one of the reviewers points out, one could just randomly
replace words (or maybe just rare words) with tags. Why
would that not work as well as curriculum learning?

For this reason, I see the "POS abstraction" as the key
contribution. But, as pointed out above, this contribution
is limited since a similar type of abstraction in a similar
experimental setup was already proposed and evaluated by Bai
et al.

Finally, instead of the simpler idea of replacing words with
POS tags, they actually do a more complex version: they use
a complete parse of the sentence and then replace phrases
with tags, as opposed to words. Unfortunately, there is no
comparison to the baseline of just replacing words. So we
have no evidence allowing us to tell whether the more
complex parsing-based setup makes a contribution.


**Summary Of Ac-Reviewer Meeting:**

We went over a draft of the metareview. Numerous changes to the draft were made based on the discussion.

Unfortunately, only the two reviewers who rated the paper a 5 were able to participate. The other two reviewers (who rated the paper a 6) might have influenced the discussion and the metareview in favor of the paper.